# Effect of pH on the Redox and Sorption Properties of Native and Phosphorylated Starches

**DOI:** 10.3390/molecules27185981

**Published:** 2022-09-14

**Authors:** Krystyna Dyrek, Elżbieta Wenda, Ewa Bidzińska, Krzysztof Kruczała

**Affiliations:** Faculty of Chemistry, Jagiellonian University in Krakow, Gronostajowa 2, 30-387 Krakow, Poland

**Keywords:** starch hydrolysis, biosorbent, chromium ions sorption, starch modification

## Abstract

Starch is a common biopolymer that can be used for removing heavy metal ions from aqueous solutions. A valuable property of starch is its functional diversity, which can be enhanced by chemical modification. Hydroxyl groups enclosed in the starch and formed during hydrolysis act as reducing agents of Cr(VI). The sorption properties of native starch depend mainly on the presence of carboxyl groups formed during redox processes and basic centers created during acid hydrolysis, while the superiority of phosphorylated starch is related to the presence of phosphate groups binding Cr(III) ions. The effectiveness of starch depends on a series of equilibria established in its aqueous suspension and chromate ions solution, where the pH is the driving force for these processes. In this article, a systematic discussion of pH changes being the consequence of chemical reactions unraveling the extraordinary functionalities of starch was given. It also explained the influence of establishing equilibria and chemical modifications of starch on the efficiency of chromium ion removal. This allowed for the development of a comprehensive mechanism for the interaction of Cr(VI) and Cr(III) ions with native and phosphorylated starch.

## 1. Introduction

Potato starch, of which Poland is one of the world’s largest producers, is an attractive sorbent because of its low cost and ease of modification. The main advantage of starch is its availability: corn, potato, rice, tapioca, wheat legume, and green fruit are common sources of commercially provided starch [1,2]. The main components of native starch are amylopectin (~70–80%), highly branched chains connected by α-(1–6) linkage, and amylose (~20–30%) built from mostly linear α-(1–4) D-glucopyranosyl molecules [3,4,5]. The exact structure and composition of the starch granules depend on many factors such as its origination from different plant sources and environmental aspects [6]. Due to the different proportions and structures of amylose and amylopectin, the properties of native starch may vary; however, regardless of the source, this biopolymer finds application in many branches of industry [7]. It is one of the essential polymers used daily in food industry applications [7,8,9]. Its physical modifications by heat, high pressure, radiation, or water are commonly accepted because of the assumption that such modified starch does not contain by-products of chemical reagents [8,10]. However, even that type of processing significantly modifies the structure and properties of the native polymer and might lead to the creation of short- and long-living radicals [11,12,13]. 

Chemical modification of native starch is achieved through etherification, esterification, cross-linking, grafting of starch, or decomposition by hydrolysis and oxidization [14,15,16,17]. The recent review papers describe the starch structure, its extraction, and modification, as well as the possible application in food and not food industries, such as nanofillers, intelligent pH-sensitive films, biofoams, aerogels of various types, bioplastics, and others [18,19]. 

One of the potential applications is the use of this native and modified biopolymer for removing transition metal ions from wastewater [20,21,22,23,24]. Environmental protection is one of the most important problems for every modern society today since the rapid development of industry has caused the increased discarding of pollutants into our surroundings. There are serious concerns about water pollution by heavy metals, organic substances, consumer care products, and endocrine compounds. Chromium(VI) is one of the most dangerous environmental pollutants for many reasons. It is contained in wastewater from numerous industries, including metallurgical, tanning, dyeing, and chemical industries [25]. It poses a serious health threat to all living organisms due to its high toxicity, i.e., carcinogenic, and mobility-related high solubility in water. The currently used methods of removing Cr(VI) from industrial wastewater are often based on sorption methods [26]. 

### 1.1. Results of Our Group in the Field of Removal Cr(VI) from Aqueous Solutions 

In two recently published works [24,27], we documented that aqueous suspensions of native and phosphorylated starch in an acidic aqueous medium are effective reducers at pH = 1 of Cr_2_O_7_^2−^ anions to Cr^3+^ and Cr^5+^ ions, which were subsequently adsorbed on starch. In native starch, the positive zeta potential on the surface of the starch grains facilitated the interaction of anionic Cr_2_O_7_^2−^ species with reducing -OH groups of the starch. Simultaneously with metal reduction, carboxyl groups of the starch were created, which served as adsorption centers for reduced chromium ions. The activation energy, calculated from kinetic data, indicated that the starch–chromium interaction was controlled by a chemical reaction formulated by D. Park as an adsorption-coupled reduction process [22]. It was evidenced for the first time, that the Cr(III) ions, generated *in statu nascendi* by reduction of Cr(VI), are much more active in coordinating to starch functional groups than those ions delivered by dissolving of the chromium(III) salts in water. In the paper [27], we present that starch phosphorylation, the more common way of starch modification [28], leads to exceptionally high efficiency of this process. The decrease in Cr(VI) ion content was similar to that of native potato starch at 45 °C. Agreement of the quantitative EPR determinations of the amount of reduced chromium with the analytical data proved that the increased amount of reduction product—in comparison to native starch—corresponds exactly to the number of PO_4_^3−^ phosphorus groups introduced into the starch structure. Therefore, the reduced Cr(III) ions were more rapidly depleted from the suspension of phosphorylated starch than from the suspension of native starch. In this way, the phosphorylated starch extorted further reduction of Cr(VI). However, it must be pointed out that during starch chemical treatment, in addition to the main reaction, hydrolysis processes take place, modifying the biopolymer properties. The mechanism of starch hydrolysis strongly depends on the pH of the solution. In the alkaline environment, the final products of hydrolysis are lower fatty acids [29], while in the acidic environment, alkaline centers are formed, influencing the changes in the pH of the solution. In both of these environments, the reduction of heavy metal ions can take place.

### 1.2. Aim of the Present Work

The current work aims to provide additional data, shedding light on the mechanism of interaction of chromium ions with starch in an aqueous medium of different pH values. The effect of the environment on the hydrolysis processes was examined in native and phosphorylated potato starch by following the changes in pH values over time. The chemical changes occurring in starch biopolymer during this process were correlated with the effectiveness of chromium anion sorption and reduction. The role additional reagents play, i.e., chromium ions on various oxidation states, was also considered. It was shown that simple experiments such as pH value measurements as a function of time allow for following reaction mechanisms. Additionally, it was documented that the well-known and described process of starch hydrolysis depends not only on pH but also on the presence of other components and starch modification. The formation of basic centers in starch in an acidic medium was proved, whereas in the basic medium, hydronium ions liberated during hydrolysis of starch participate in the reduction of Cr(VI).

## 2. Results and Discussion

### 2.1. Changes of pH in Water Suspensions of Starch as a Function of Time

#### 2.1.1. Native Starch in Water 

Figure 1 presents changes in pH in the water suspension of native starch as a function of time. In the case of the initial value of pH = 7, no pH changes occurred during 24 h, which proves the stability of the system. For the initial value of pH ≠ 7, in two cases: pH = 4 and pH = 11, significant variations in pH values were observed.

In an acidic medium (pH = 4), the pH rapidly increases to the value of 5.8, which means that the concentration of H_3_O^+^ ions decreases almost 100 times. After 24 h. the pH of the solution stabilizes with a value equal to 6.1. Changes in pH are a consequence of starch hydrolysis [29], which, in an acidic medium, consists of the breaking of some α (1–4) and/or α (1–6) glucosidic bonds of amylose and amylopectin with the formation of negatively charged basic centers. These centers attract H_3_O^+^ ions from the water, which leads to a lowering of the acidity of the solution and creates positive potential ζ of the surface of native starch grains, as can be seen in Figure 2. In more acidic solutions (pH = 1 and pH = 2), the same process occurs, however, due to the high total number of hydronium ions, the effect of the pH changes is less conspicuous. It should be expected that hydrolysis occurs mainly in the amorphous part of the starch structure [30]. 

Significant changes in pH value occur also in the suspension with an initial pH value equal to pH = 11. During the first hour of contact time, the value of pH decreases to pH = 7.6, which indicates a significant increase in the number of hydronium ions in the solution.

In basic solutions, hydrolysis leads to the breaking of glucosidic rings of starch and the creation of lower organic acids, which dissociate with the production of H_3_O^+^ ions [29], resulting in an effective reduction of the pH value. 

#### 2.1.2. Phosphorylated Starch in Water

In Figure 3, variations of pH values occurring in water suspensions of the phosphorylated starch as a function of time is presented. Similar changes in the pH value occur and became stabilized as previously observed for native starch (Figure 1) despite using a 5-fold less amount of modified starch. The most significant effects occur in the case of initial values of pH = 4 and pH = 11. In an acidic medium, the stabilized pH value for phosphorylated starch is equal to 6.1, which is the same as for suspension of native starch in water.

It should be seen that the results of measurements of ζ potential in an acidic medium for the phosphorylated starch differ from those obtained for native starch (Figure 2). The surface of the modified starch is neutral [27], whereas native starch reveals a positive value of the ζ potential [24]. This difference indicates stronger interaction of basic centers in modified starch with positive hydronium ions, compared to those produced in native starch.

Chemical modification of the starch by phosphorylation increases the effectiveness of the hydrolysis in the basic medium, which results in a little change in initial pH value (∆pH = 3.5) as compared with native starch (∆pH = 3.4). During basic hydrolysis, more hydronium ions are created in phosphorylated starch occurring under the same conditions as for native starch.

### 2.2. Changes in pH in an Aqueous Solution of K_2_Cr_2_O_7_


The results of the pH measurements in an aqueous solution of K_2_Cr_2_O_7_ as a function of time are presented in Figure 4. No changes in initial pH values with pH = 1, 4, and 7 are observed during the first 24 h, which indicates the stability of these systems. Significant changes occur only for the solution with an initial pH = 11. This value changes slowly, and after 4 h is equal to 8.6.

The concentration of Cr(VI) = 1 g/L, room temperature.

In a water solution of K_2_Cr_2_O_7_, hydrolysis occurs according to the reaction described by the Equation (1):Cr_2_O_7_^2−^ + 3H_2_O = 2CrO_4_^2−^ + 2H_3_O^+^(1)

In a basic medium (pH = 11), the equilibrium is established with CrO_4_^2−^ as the dominating form of Cr(VI) concentration.

Hydronium ions liberated by hydrolysis of potassium dichromate in a basic medium cause a lowering of the initial pH value of the K_2_Cr_2_O_7_ solution. 

### 2.3. Changes of pH in the Suspensions of Starch in an Aqueous Solution of K_2_Cr_2_O_7_ as a Function of Time

#### 2.3.1. Native Starch in an Aqueous Solution of K_2_Cr_2_O_7_

As can be seen in Figure 5, the pH value of the aqueous solution of K_2_Cr_2_O_7_ in the presence of native starch changes, which is associated with the hydrolysis of the starch and processes induced by the third component—Cr(VI) ions.

In an acidic medium (pH = 4), basic centers with a negative charge, created during hydrolysis, interact with positively charged H_3_O^+^ ions from the solution. This is responsible for the positive potential of the starch surface and facilitates the approaching of Cr_2_O_7_^2−^ anions to starch [24]. 

The Cr(VI) species became reduced mostly to Cr(III) ions according to the reaction described by Equation (2): Cr_2_O_7_^2−^ + 14 H_3_O^+^ + 6 e^−^ = 2 Cr^3+^ + 21 H_2_O →E° = 1.33 V(2)

However, Cr(V) ions in smaller amounts were also created [24]. Simultaneously with the reduction of Cr(VI), a corresponding number of OH^−^ groups of the starch are oxidized to very active sorption centers -COO^−^. Freshly reduced Cr(III) cations are bonded to starch by these new adsorption sites. Additionally, a certain number of reduced Cr(III) ions are blocking the basic centers of the starch created by acidic hydrolysis, because they compete efficiently in the occupation of these negative sites with the hydronium ions due to their higher charge than the +1 positive charge of H_3_O^+^. As a result of the blockade of basic centers, a certain number of H_3_O^+^ ions persist in the solution such that the value of ∆pH = 1.9 in the acidic solution of potassium dichromate is smaller than that of native starch suspended in water (∆pH = 2.1, Figure 1 and Table 1).

In the basic medium of native starch in a water solution of K_2_Cr_2_O_7_, the pH value decreases to 8.4. This change (∆pH = 2.6) is smaller than that observed for suspension of native starch in pure water (∆pH = 3.4) (Table 1). The difference is caused by the reduction of Cr(VI) ions by starch and consumption of H_3_O^+^ ions in the redox reaction, according to reaction (2). The presence of Cr(III) ions in starch was confirmed by EPR data (Section 2.5). Hydronium ions created by the starch hydrolysis in the basic medium are also partially consumed in reaction (2), resulting in a lowering of the pH change. 

In a basic medium the following reaction is also possible:CrO_4_^2−^ + 4 H_2_O + 3e^−^ = Cr(OH)_3_ + 5 OH^−^ E° = −0.13 V (3)

However, due to the significant difference in E° values, reaction (2) is privileged.

The third component (Cr(VI)) influences pH values in the water-starch system.

(1)Freshly reduced Cr(III) ions blockade the basic centers formed during acidic hydrolysis. As a result, a certain amount of hydronium ions remains free in the solution and may participate in the reduction of Cr(VI) in an acidic medium. Consumption of these ions in reaction (2) is a reason for a smaller ∆pH value (∆pH = 1.9) than that for native starch in water (∆pH = 2.1). Diversity of sorption centers increases because basic centers efficiently play the role of sorbents and quickly remove the reduction products (Cr(III)), stimulating in this way the further reduction of Cr(VI).(2)Changes in pH in a basic medium are smaller (∆pH = 2.6) than in pure water (∆pH = 3.4) because a part of hydronium ions formed during the basic hydrolysis of starch is consumed in the reduction of a certain amount of Cr(VI) ions.

#### 2.3.2. Phosphorylated Starch in an Aqueous Solution of K_2_Cr_2_O_7_

In an acidic medium, in the case of phosphorylated starch and in the presence of Cr(VI) ions, pH value increases less (∆pH = 1.4 in Figure 6) than that in the suspension of phosphorylated starch in water (∆pH = 2.1, Figure 3), which indicates that basic centers are partially blocked by Cr(III) ions, whereas H_3_O^+^ ions remain in solution. Comparing this effect (∆pH = 1.4) with that found for native starch (∆pH = 1.9, Figure 5), in the presence of Cr(VI) ions in an acidic medium, leads to the conclusion that modification of the starch caused a greater number of Cr(III) ions to block the basic centers in phosphorylated than that in native starch.

In a basic medium, in the presence of Cr(VI) ions, the decrease in ∆pH value of phosphorylated starch is also smaller (∆pH = 3.1, Figure 6) than in pure water (∆pH = 3.5, Figure 3), which indicates that hydronium ions created by the starch hydrolysis in a basic medium are partially consumed by reduction of Cr(VI) ions (reaction (2)), resulting in less significant pH change. Comparing this effect (∆pH = 3.1, Figure 6) with that for native starch (∆pH = 2.6, Figure 5), one can conclude that in modified starch the process of Cr(VI) reduction engaged a smaller number of hydronium ions than in native starch. This effect suggests that a different reduction process is involved in non-modified and phosphorylated starches, which are in line with quantitative EPR measurements (Section 2.5). Indeed, EPR data indicate that the ratio of reduction products (Cr(V)/Cr(III) ions) in phosphorylated starch increases much more significantly in phosphorylated than in native starch with an increasing value of pH (*vide infra*). 

The effect of the third component (Cr(VI) ions) in the phosphorylated starch–water system consists of:(1)The basic centers created by acidic hydrolysis play a role in especially active sorption centers of the reduction product (Cr(III)) in phosphorylated starch. In an acidic medium, blockade of basic centers by Cr(III) ions occurs and H_3_O^+^ ions remain in the solution.(2)The ratio of reduction products (Cr(V)/Cr(III) ions) increases in phosphorylated starch much more significantly than in native starch with an increasing value of pH.

### 2.4. Changes of pH in Suspensions of Native and Phosphorylated Starch in Water Solutions of Cr(NO_3_)_3_·9(H_2_O)

Chromium(III) in water solutions exists in the form of aqua complexes (Equation (4)) and aqua hydroxo complexes (Equation (5)) formed by hydrolysis. Hydrolysis leads to an acidic medium of the solutions:Cr(NO_3_)_3_ + nH_2_O → [Cr(H_2_O)_n_]^3+^
(4)
[Cr(H_2_O)_n_]^3+^ + H_2_O → [Cr(OH)(H_2_O)_n−1_]^2+^ + H_3_O^+^
(5)

Comparing the results obtained for native starch in water (Figure 1) with pH changes occurring in the suspension of starch in water solution of Cr(NO_3_)_3_, presented in Figure 7, no changes were observed for Cr(III) concentration equal to 1 g/L. It points out the complete blockade of basic centers by Cr(III) ions added to the starch suspension. No hydronium ions are adsorbed on these centers, and the value of pH does not change. However, for lower Cr(III) concentration (0.1/L of Cr(III)/L), an increase in pH to 4.6 (∆pH = 0.6) was detected since lower Cr(III) ion concentration causes that blockade to be less efficient. Similar behavior was observed in the case of phosphorylated starch. 

These results can be rationalized by the assumption that for higher concentrations of Cr(III), products of hydrolysis (reactions 4 and 5) undergo sorption on basic centers of the starch, thereby blocking sorption of H_3_O^+^ ions and making changes in pH values less noticeable.

### 2.5. Quantitative EPR

#### 2.5.1. Effect of Contact of Cr(VI) and Cr(III) Ions with Hydrolyzed Potato Starch 

In Table 2, the results concerning the effectiveness of the reduction of Cr(VI) ions in aqueous solutions at pH = 1 and sorption of reduced Cr(III) ions are presented for native potato starch and the samples of native starch pretreated by hydrolysis in an acidic and basic medium. To separate information concerning two functions of the starch—as a reducer of Cr(VI) and adsorbent of reduced chromium ions—two kinds of Cr(III) ions were examined: (1) produced by freshly reduced (Cr_2_O_7_)^2−^ ions and (2) provided from Cr(NO_3_)_3_ ·9H_2_O solution. In the first case the role of starch was double: to reduce Cr(VI) and adsorb the reduced ions on the polymer, whereas in the second case the starch plays only the role of adsorbent.

The values concerning native starch and starch after pretreatment in acidic conditions are similar. In both cases, the results obtained for Cr(III) ions produced by freshly reduced Cr_2_O_7_^2−^ ions are higher than those for the ions provided by Cr(NO_3_)_3_·9H_2_O compound. In the salt solution, the Cr(III) ions form hexaaqua complexes in which central Cr(III) cation is surrounded by H_2_O ligands shielding it from the sorption centers of the starch. As a consequence, the sorption is impeded. On the other hand, Cr(III) ions produced by in situ reductions of Cr_2_O_7_^2−^ ions exhibit an unoccupied coordination sphere and therefore may easily approach the sorption centers. 

Hydrolysis in an acidic medium is a very rapid process and progresses to some extent, therefore the content of Cr(III) adsorbed on native starch at the applied conditions (pH = 1) is practically equal to that on starch pretreated for 24 h in an acid solution. 

Starch hydrolyzed in a basic environment reveals higher effectiveness of sorption than both native starch and starch hydrolyzed in an acidic medium. In the basic medium, opening of the glucosidic rings takes place [29], which leads to the creation of new, very active sorption centers in the form of carboxyl groups. This fact is responsible for significant improvement of the sorption capacity after hydrolysis of the polymer in basic conditions. More significant improvement of the sorption in the case of Cr(III) ions supplied from salt solution than from freshly reduced Cr(IV) ions indicates that the effect is related predominantly to sorption capacity and less to reduction efficiency. 

Pretreatment of the starch in a basic medium increases its effectiveness in further contact with Cr(VI). Sorption properties are especially enhanced.

#### 2.5.2. Effect of pH on the Amount of Cr(III) and Cr(V) Ions Adsorbed on the Surface of the Native and Phosphorylated Potato Starch

All the data reported in our previous papers were obtained in aqueous solutions with a pH value equal to 1, favorable for the reduction of Cr(VI), introduced into this system in the form of Cr_2_O_7_^2−^ ions, to Cr(III) ions, followed by adsorption of them on the starch. The aim of the experiment presented below was to investigate the interaction of starch with K_2_Cr_2_O_7_ in a range of pH values pH = 1–12 and to check how the pH value influences the reduction mechanism. After a certain time, the starch-containing reduced chromium was separated, and the content of paramagnetic chromium ions (Cr(III) and Cr(V)) in starch was determined by quantitative EPR spectroscopy.

Data presented in Figure 8 indicate, that with increasing pH value in both starches, content of Cr(III) decreased, whereas the content of Cr(V) increased. It means that pH values influence the way of chromium reduction with the transfer of three (reduction of Cr(VI) to Cr(III)) or only one electron (reduction of Cr(VI) to Cr(V)) per one Cr(VI) atom. It indicates that the main parameter determining the mechanism of Cr(VI) reduction is still the value of pH, but the influence on this process is also modification of the starch.

In native starch suspended in Cr(VI) solution, -COO^−^ sorption centers created proportionally to the quantity of reduced Cr(III) and/or Cr(V) ions and basic centers created by acidic hydrolysis are present. In phosphorylated starch, there are additionally -PO_4_^3−^ groups that act as very active sorption centers. In a basic medium, initial value of pH is more strongly reduced in phosphorylated (Figure 3) than in native starch (Figure 1). The mechanism of reduction of Cr(VI) ions changes under influence of increasing pH values.

## 3. Materials and Methods

**Materials:** Commercial food-grade potato starch was obtained from WPPZ SA, Poland. Analytical grade solutions 1 M of HCl and NaOH were supplied by Chempur, whereas K_2_Cr_2_O_7_ and chromium(III)nitrate (Cr(NO_3_)3·9H_2_O) were delivered by POCh, Poland. 

**Starch Phosphorylation:** The procedure of phosphorylation was based on papers Lim and Seib [31], Sang and Seib [32], and Bidzińska et al. [27]. In a typical experiment, monostarch phosphate was prepared from native potato starch introduced to an acidic aqueous solution (pH = 6) containing sodium trimetaphosphate/sodium tripolyphosphate (STMP/STPP, 99/1, *w*/*w*) and Na_2_SO_4_. After mixing this suspension for 1 h at room temperature, it was dried at 45–50 °C until the water content decreased below 20%. The solid residue was heated at 130 °C for 2 h, ground, and mixed with NaOH solution (pH = 6.5) in distilled water. This treatment was repeated three times, then the product was dried at 50 °C overnight.

**Analyses:** Phosphorus content in the starch phosphate was determined according to Polish Norm [33]. In the first step, the mineralization of solid samples in a mixture of concentrated sulphuric(VI) and nitric(V) acids was performed. Dissolved phosphates were transformed into phosphoromolibdenic(VI) acid, which was next reduced with ascorbic acid. Phosphorus content in the intensive blue solution of this compound was determined by spectrophotometric analysis at a wavelength of 680 nm. Monostarch phosphate used in the above-described experiments contained 0.33 wt % of phosphorus.

**Starch hydrolysis in acidic medium:** The suspension of 10 g of native starch in 45 cm^3^ water with pH = 1 (obtained with 1 M HCl) was shaken for 1 h. The pH value of the solution was measured, and the starch was washed on a Büchner funnel with distilled water until a negative reaction for Cl^−^ ions was reached. The solid sample was dried for 24 h at 50 °C.

**Starch hydrolysis in basic medium:** The pH of the 40% weight suspension of the starch in water was adjusted to the value equal to 10. The content was mixed for 50 min at room temperature. The pH was then lowered to 7, and the solid was filtered off and washed with distilled water. The sample was dried for 24 h at 50 °C.

**Quantitative EPR spectroscopy:** To prepare samples for EPR measurements, 5 g of the starch pretreated in the above-mentioned way were contacted with 22.5 cm^3^ Cr(VI) or Cr(III) solutions with a concentration of 5 g Cr/L at pH = 1 and mixed for 24 h at room temperature. Then, the samples were filtered and dried for 24 h at 50 °C. EPR spectra were measured at room temperature by an Elexsys E-500 spectrometer (Bruker, Billerica, MA, USA) at X band (9.8 GHz) with a modulation amplitude of 0.1 mT and microwave power of 3 mW. The number of paramagnetic chromium ions was determined by comparison of the value of the double integral of the area of EPR signals of an appropriate sample with that of the EPR standard [34].

**The procedure of pH measurements:** The pH values, in the range 1–12, were obtained by adding the required amount of HCl and NaOH solutions to volumetric flasks filled with 22.5 cm^3^ of distilled water, K_2_Cr_2_O_7,_ or Cr(NO_3_)_3_ solution containing 1 g of Cr/L. Then, either 5.00 g of native potato starch or 1.00 g of phosphorylated starch was added, and the suspensions were mixed at room temperature for 24 h to properly shake up the contents; apparatus from the firm Elpin was used. Every hour, the pH value of suspensions was measured by an Elmetron pH-meter equipped with a glass electrode from the firm Eurosensor. Measurements were performed at room temperature, with a precision of reading equal to 0.05.

**Zeta Potential Measurements:** Zeta potentials were determined in the range of pH values from 1 to 10 by Zetasizer Nano S combined with the MPT-2 Autotitrator (Malvern Instruments, Malvern, U.K.).

## 4. Conclusions

Native starch in a neutral water suspension at ambient temperature is reasonably stable; however, it undergoes hydrolysis in contact with water of a pH value different than 7. The mechanism of starch hydrolysis in water suspension depends not only on pH but also on the presence of other components and starch modification. The mechanism can be followed by such simple experiments as suspension of starch in water or in solutions of chromium(VI) and chromium(III) with pH value measurements as a function of time.

In acidic solutions, the breaking of α-1,4- and α-1,6- bonds of native starch creates basic centers. The interaction of these negatively charged centers with hydronium ions present in the slurry leads to the attraction of H_3_O^+^ ions and the creation of the positive ζ potential of the native starch surface, whereas the ζ potential of phosphorylated starch is neutral. This difference indicates stronger interaction between hydronium ions and basic centers created by acidic hydrolysis in phosphorylated starch than with similar centers formed in native starch.

In basic solutions (pH = 11), hydrolysis leads to the breaking of glucosidic rings of starch and the creation of lower organic acids, which dissociate with the production of H_3_O^+^ ions. The effect of hydrolysis in a basic medium results in an effective lowering of pH value. More hydronium ions are created under the same conditions in phosphorylated than in native starch.

The third component, namely Cr(VI) ions, influences changes in pH values of hydrolyzed starch by two mechanisms: BLOCADE in an acidic medium and REDUCTION in a basic medium. Negatively charged basic centers created in starch by acidic hydrolysis are blocked by Cr(III) ions, which are products of Cr(VI) reduction. Simultaneously, a suitable quantity of -COO^−^ groups are formed. Both of them are negative and consist of attractive sorption centers for freshly reduced Cr(III) ions. In modified starch, additionally, the third sorption center exists, namely a very active sorption center incorporated into the structure of the -PO_4_^3−^ group, which not only immediately attracts Cr(III) ions, but also creates much more effective basic centers during acidic hydrolysis.

In the basic medium, part of hydronium ions formed during the basic hydrolysis of starch are consumed in the reduction of a certain amount of Cr(VI) ions. Simultaneously, an appropriate quantity of -COO^−^ groups are formed. They have a negative charge; therefore, they are attractive sorption centers for freshly reduced chromium ions. The higher pH causes a change in the reduction mechanism comprised of lowering the content of Cr(III) and increasing the number of Cr(V) ions upon increasing the pH value of the solution.

## Figures and Tables

**Figure 1 molecules-27-05981-f001:**
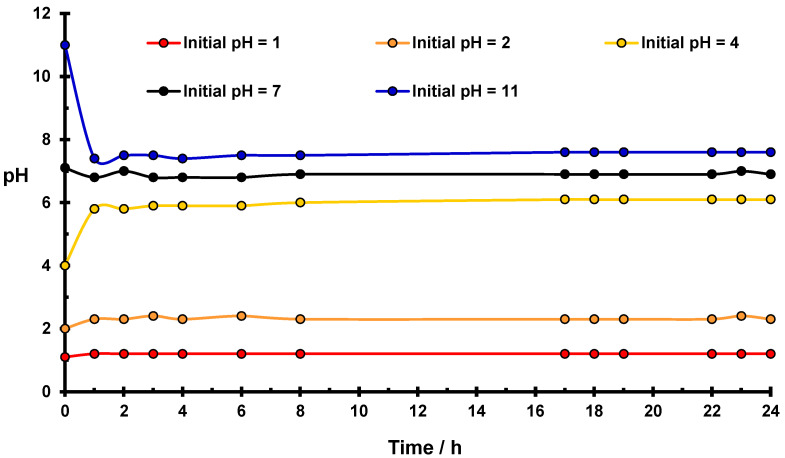
Changes of pH for the native starch suspension in water as a function of time. Conditions: 5 g of starch in 22.5 cm^3^ of water at room temperature.

**Figure 2 molecules-27-05981-f002:**
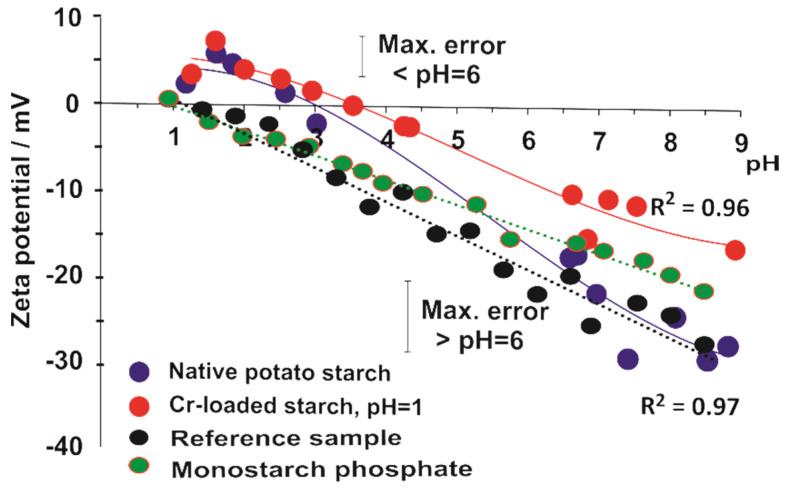
The effect of pH on the zeta potential of native and modified potato starch.

**Figure 3 molecules-27-05981-f003:**
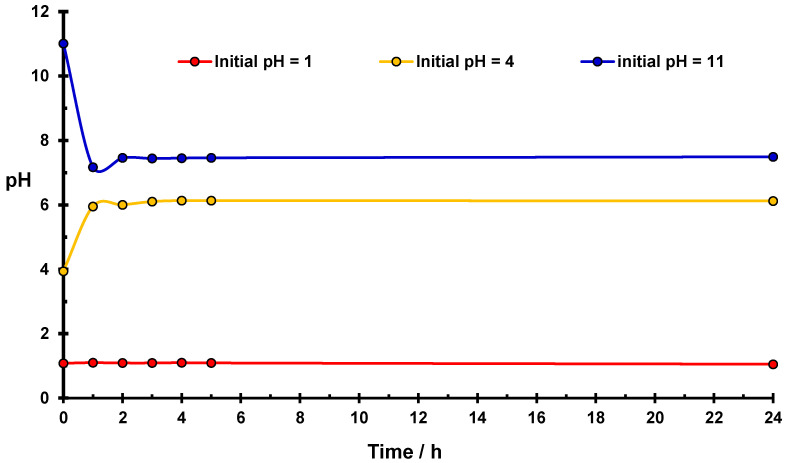
Changes of pH for the starch suspension in water as a function of time. Conditions: 1 g of the phosphorylated starch in 22.5 cm^3^ of water at room temperature.

**Figure 4 molecules-27-05981-f004:**
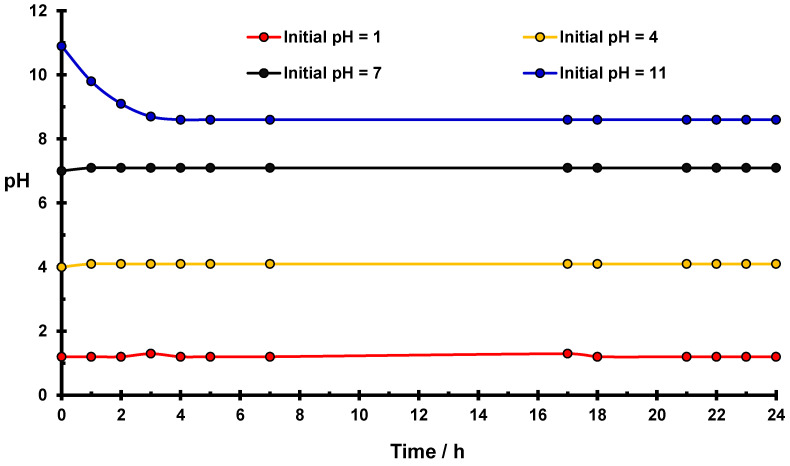
Changes of pH for the water solution of K_2_Cr_2_O_7_ as a function of time.

**Figure 5 molecules-27-05981-f005:**
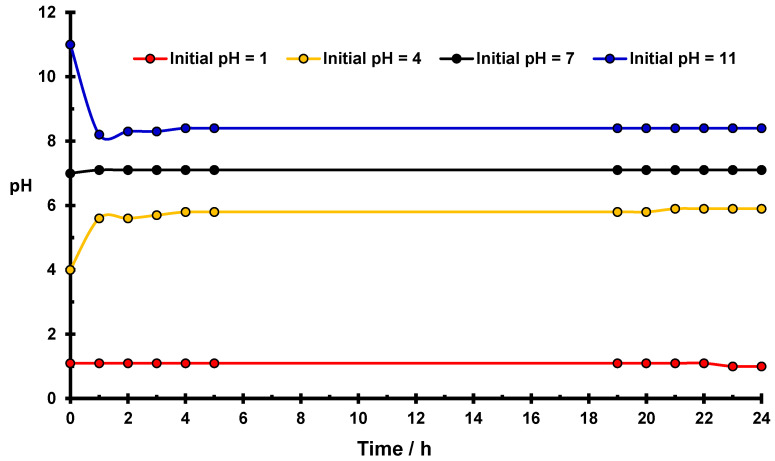
Changes of pH for the native starch suspension in water solution of K_2_Cr_2_O_7_ (concentration of Cr(VI) = 1 g/L) as a function of time. Conditions: 5 g of starch in 22.5 cm^3^ of a solution, room temperature.

**Figure 6 molecules-27-05981-f006:**
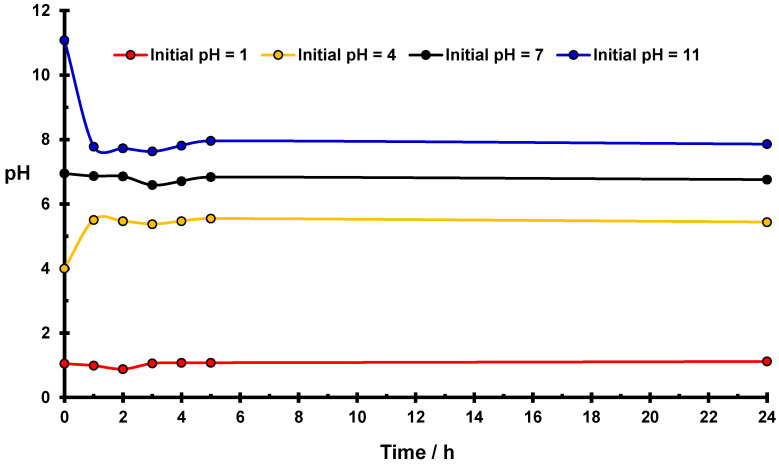
Changes in pH for the phosphorylated starch suspension in water solution of K_2_Cr_2_O_7_. Conditions: concentration of Cr(VI) = 1 g/L, 1 g of starch, 22.5 cm^3^ of a solution, room temperature.

**Figure 7 molecules-27-05981-f007:**
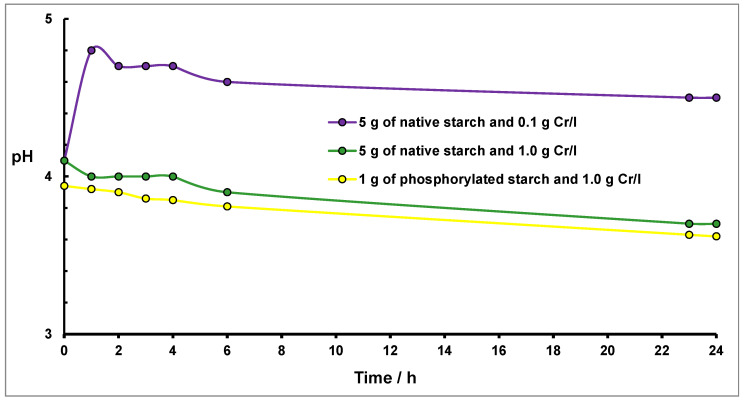
Changes of pH for the native and phosphorylated starch suspensions in the Cr(NO_3_)_3_ solutions as a function of time. Conditions: 5 g of native starch in 22.5 cm^3^ of 0.1 g Cr/L (purple line), 1.0 g Cr/L (green line) Cr(NO_3_)_3_ solutions, and 1 g of phosphorylated starch in 22.5 cm^3^ of 1.0 g Cr/L Cr(NO_3_)_3_ solution (yellow line), room temperature.

**Figure 8 molecules-27-05981-f008:**
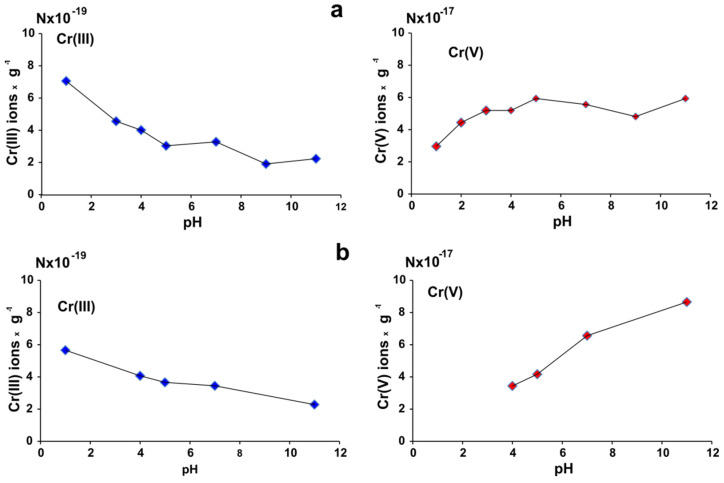
Number of Cr(III) and Cr(V) ions adsorbed on native potato starch (**a**) and on monostarch phosphate (**b**) with 0.33 wt% P as a function of pH value of the solution. Experimental conditions: for native starch initial Cr(VI) concentration = 5 g/L, volume = 22.5 mL, starch amount = 5 g, contact time = 24 h.; for phosphorylated starch: initial Cr(VI) concentration = 1 g/L, volume = 22.5 mL, starch amount = 1 g, contact time = 24 h, room temperature.

**Table 1 molecules-27-05981-t001:** Collected data from Section 2.1, Section 2.2, Section 2.3 and Section 2.4 concerning pH changes in water suspensions of the native and phosphorylated starch as a function of time.

	Native Starch		Phosphorylated Starch		No Starch	
	Acidic Medium	Basic Medium	Acidic Medium	Basic Medium	Acidic Medium	Basic Medium
**Pure Water**	Figure 1		Figure 3			
	4 → 6.1 ∆pH = 2.1	11 → 7.6 ∆pH = 3.4	4 → 6.1 ∆pH = 2.1	11 → 7.5 ∆pH = 3.5	–	–
**H_2_O + K_2_Cr_2_O_7_**	Figure 5		Figure 6		Figure 4	
	4 → 5.9 ∆pH = 1.9	11 → 8.4 ∆pH = 2.6	4 → 5.4 ∆pH = 1.4	11 → 7.9 ∆pH = 3.1	–	11 → 8.6 ∆pH = 2.4

**Table 2 molecules-27-05981-t002:** Effect of contact of Cr(VI) and Cr(III) ions in aqueous solutions (pH = 1) with native and hydrolyzed potato starch. Conditions: concentration of Cr 5 g/L, 22.5 cm^3^ of a solution, 5 g of starch, contact time 24 h, room temperature.

Type of Starch	Initial State of Chromium	Number of Cr^3+^ Ions Adsorbed on 1 g of Starch *
Native	Cr(VI) from K_2_Cr_2_O_7_	7.35 × 10^19^
Hydrolyzed in an acidic medium		7.90 × 10^19^
Hydrolyzed in a basic medium		8.66 × 10^19^
Native	Cr(III) from Cr(NO_3_)	4.72 × 10^19^
Hydrolyzed in an acidic medium		4.35 × 10^19^
Hydrolyzed in a basic medium		8.39 × 10^19^

***** Mean value of three measurements.

## Data Availability

The data presented in this study are available on request from the corresponding author.

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
