# Peer review of "Effect of pH on the Redox and Sorption Properties of Native and Phosphorylated Starches"

_molecules, 2022, doi:10.3390/molecules27185981_

Round 1

Reviewer 1 Report (Previous Reviewer 2)

I am currently fine with the current manuscript version. Authors properly addressed my original comments.

Reviewer 2 Report (Previous Reviewer 1)

The paper can be accepted.

This manuscript is a resubmission of an earlier submission. The following is a list of the peer review reports and author responses from that submission.

Round 1

Reviewer 1 Report

Dyrek et al. studied the effect of pH on the redox and sorption properties of native and phosphorylated starches. The manuscript reported some interesting results. However, here are many shortcomings:

  1. The Introductionpart should be written in several paragraphs rather than a long paragraph.
  2. The Introductionpart: The novelty of this paper needs to be further summarized.
  3. Abstract sections are usually written in the past tense. Because the experiment was done in the past.
  4. Introduction section does not show the comprehensive research progress of the starches. The background knowledge on starchesneeds to be further reviewed, by consulting the recently published articles, for example: Molecules 2022, 27(7), 2119; Food Hydrocolloids Volume 117, August 2021, 106690; Food Chemistry Volume 342, 16 April 2021, 128325; Starch,Volume73, Issue3-4, March 2021,2000013;
  5. In addition, other relevant literature about the starches published in molecules journals should also be added in the Introduction section.
  6. Figure 1,3,4,5,7,8, There is something wrong with the figures. Because the reader doesn't know which line corresponds to what sample. Please revise them.
  7. Some of the results in the article lack detailed explanation.
  8. References: The format of references should be uniform.
  9. Moreover, Other small issues along the manuscript resulting from a less correct use of the English language should also be considered.

From the above, the authors should perform a deep revision of the manuscript taking into consideration at least the comments above. 

Reviewer 2 Report

I have read the paper and the authors can find below my comments:

  •  authors did not frame their research question compared to an extant review on the topic, hindering the reader to detect the novelty of the authors' work. 
  • discussion on the feasibility of the method the authors proposed is missed. Data on costs and barriers to the industrial use of the method proposed would enrich the paper. 
  • the methodology section needs to be discussed deeper, adding further details, as now is just broadly described.
  • the paper ends without a conclusion section where authors should simultaneously describe what have discovered, and exploit it to add something new to the current works. Also, authors have to discuss the possible limitations of their study or the insights for future directions of research.

Thus I suggest rejection for this paper.